# Comparison of Active Bone Conduction Hearing Implant Systems in Unilateral and Bilateral Conductive or Mixed Hearing Loss

**DOI:** 10.3390/brainsci13081150

**Published:** 2023-07-31

**Authors:** Andrea Canale, Anastasia Urbanelli, Maria Gragnano, Valerio Bordino, Andrea Albera

**Affiliations:** 1ENT Unit, Department of Surgical Sciences, University of Turin, 10126 Turin, Italy; anastasia.urbanelli@unito.it (A.U.); mariagragnano91@gmail.com (M.G.); aalbera@hotmail.com (A.A.); 2Department of Public Health Sciences and Paediatrics, University of Turin, 10126 Turin, Italy; valerio.bordino@unito.it

**Keywords:** bilateral conductive hearing loss, unilateral conductive hearing loss, bone conduction implant, binaural hearing

## Abstract

Background: To assess and compare binaural benefits and subjective satisfaction of active bone conduction implant (BCI) in patients with bilateral conductive or mixed hearing loss fitted with bilateral BCI and patients with monaural conductive hearing loss fitted with monaural BCI. Methods: ITA Matrix test was performed both on patients affected by bilateral conductive or mixed hearing loss fitted with monaural bone conduction hearing implant (Bonebridge, Med-El) before and after implantation of contralateral bone conduction hearing implant and on patients with monaural conductive or mixed hearing loss before and after implantation of monaural BCI. The Abbreviated Profile of Hearing Aid Benefit (APHAB) questionnaire was administered to both groups of subjects and the results were compared with each other. Results: Patients of group 1 reported a difference of 4.66 dB in the summation setting compared to 0.79 dB of group 2 (*p* < 0.05). In the squelch setting, group 1 showed a difference of 2.42 dB compared to 1.53 dB of group 2 (*p* = 0.85). In the head shadow setting, patients of group 1 reported a difference of 7.5 dB, compared to 4.61 dB of group 2 (*p* = 0.34). As for the APHAB questionnaire, group 1 reported a mean global score difference of 11.10% while group 2 showed a difference of −4.00%. Conclusions: Bilateral BCI in patients affected by bilateral conductive or mixed hearing loss might show more advantages in terms of sound localisation, speech perception in noise and subjective satisfaction if compared to unilateral BCI fitting in patients affected by unilateral conductive hearing impairment. This may be explained by the different individual transcranial attenuation of each subject, which might lead to different outcomes in terms of binaural hearing achievement. On the other hand, patients with unilateral conductive or mixed hearing loss fitted with monaural BCI achieved good results in terms of binaural hearing and for this reason, there is no absolute contraindication to implantation in those patients.

## 1. Introduction

Binaural rehabilitation of patients affected by conductive hearing loss has been strongly discussed over the years and is still lined with criticality about its effectiveness and benefit. In fact, as demonstrated by Stenfelt, due to the individual TA (transcranial attenuation), any BCI (bone conduction implant) stimulation through the so-called “cross hearing” not only the ipsilateral ear but also the contralateral cochlea leads to interfering in binaural cues [1].

With regard to binaural restoration of patients affected by monaural conductive hearing loss, in the past many authors tried to demonstrate binaural benefits after fitting with BCI in such patients and their results turned out to be conflicting: in some works, BCI seemed not to improve sound localisation under noise conditions [2] and was demonstrated not to gain binaural restoration in subjects with congenital unilateral hearing loss in terms of directional hearing and localisation ability [3]. On the other hand, some authors reported binaural benefits in such patients through monaural BCI. For example, Vogt et al. demonstrated that cross-stimulation of BCI in subjects with monaural conductive hearing loss did not affect sound localisation abilities [4], while Agterberg et al. reported summation advantage and improving directional hearing in patients affected by unilateral conducted hearing loss fitted with monaural BCI [5]. On the other hand, they evidenced a lack of binaural benefits in patients with congenital hearing impairment due to the influence of the critical period in the development of binaural hearing and to the crossover stimulation of contralateral cochlea that unavoidably might deteriorate binaural restoration in patients affected by monaural conductive hearing loss [5]. Other authors showed advantages in speech perception with BCI without studying the effects of binaural hearing [6,7]. In particular, Danhauer et al. demonstrated benefits with the implantation of unilateral BAHA (bone-anchored hearing aids) and reported a reduction in activity limitations when using it, but in their work, both speech and noise were presented at the frontal speaker (S_0_N_0_) and no further settings were analysed [7]. Some authors, finally, reported benefits in reducing the handicap resulting from monaural conductive hearing loss and emphasised the patients’ satisfaction derived from the use of BCI in their daily life [8].

The efficacy of binaural rehabilitation through BCI in patients affected by bilateral conductive hearing loss, on the other hand, has been strongly demonstrated over the years and found favourable consensus in many authors [9,10,11,12]. Several studies were conducted about the benefits in terms of sound localisation [13] and speech perception in noise [14] through the application of bilateral BCI in bilateral conductive hearing loss. Moreover, thanks to the summation effect, the reduction of the head shadow effect and the squelch effect, bilateral implantation of BC (bone conduction) aids in patients affected by bilateral conductive hearing loss resulted in a greater benefit in terms of binaural hearing and subjective satisfaction rather than unilateral BCI [9].

Given these assumptions, the question remains whether unilateral BCI in patients affected by monaural conductive hearing loss might provide satisfactory outcomes in terms of binaural hearing if compared to an additional BCI in patients affected by bilateral conductive hearing loss already fitted with a unilateral implant. The aim of the presented study is to demonstrate whether bilateral amplification with BCI in patients affected by bilateral conductive hearing loss could provide better results in terms of speech perception under noise conditions and subjective satisfaction compared to benefits gained by unilateral BCI in patients affected by monaural conductive hearing loss.

## 2. Materials and Methods

This study has been performed in accordance with the ethics standards laid down in the 1964 Declaration of Helsinki and informed written consent was obtained from all patients. Committee approval was taken from University of Turin (Date: 13 March 2018, Protocol Number: 0026286; CS2/622).

### 2.1. Patients

The cohort included two different groups of patients (group 1 and group 2) whose audiometric results were compared. Group 1 included 7 female patients affected by bilateral conductive or mixed hearing implanted with bilateral bone conduction hearing aid (BONEBRIDGE^TM^, Med-El, Innsbruck, Austria) in the Otolaryngology Division of Molinette Hospital, Turin. Group 2 included 7 patients (3 female and 4 male) affected by monaural conductive or mixed hearing loss who underwent surgery implantation of monaural bone conduction hearing aid (BONEBRIDGE^TM^, Med-El, Innsbruck, Austria) in the Otolaryngology Division of Molinette Hospital, Turin. The main age at implantation was 30 years (SD, 19.47 years) for group 1 and 48 years (SD, 15.09 years) for group 2. Participation in the study was voluntary. All the surgical procedures were performed by the same surgeon. Inclusion criteria for all groups were patients older than 18 affected by bilateral (group 1) or monaural (group 2) conductive or mixed hearing loss fitted with bilateral (group 1) or monaural (group 2) BCI. Exclusion criteria were age under 18 years, learning disability and attention disorders. Information for each subject in both groups including age at implantation, gender, aetiology of deafness (congenital or acquired), side of implantation and pre-implantation audiometric scores are summarised in Table 1. ISO (International Organization for Standardization) 7029:2017 normative was adopted to assess the hearing threshold deviation for audiometric tones of subjects in the study to prove that any observed hearing loss in the AC thresholds was associated with age and not with any cochlear injury [15]. According to the indications of the company house (Med-El), the retrosigmoid approach was used for all the subjects in the study. All patients underwent preoperative CT and MRI before surgery, to evaluate the individual anatomy of the skull and to exclude any bone deformity that could interfere with the implantation of the aid. All implants were activated two weeks after surgery and regular mapping of the implants was performed.

### 2.2. Intervention

All participants underwent pure tone audiometry (250–8000 Hz) to measure bilateral hearing thresholds in daily life before implantation. The audiological pattern of all groups is reported in Table 1.

Speech intelligibility in noise with the BCI was tested using ITA Matrix test [16]. Results are expressed in decibels and represent the signal-to-noise ratio (SNR) at which a subject understands 50% of the words spoken by the test administrator. Audiological evaluations of all patients were carried out between May 2021 and January 2023. ITA Matrix test was performed using two loudspeakers with a radius of 1 m in three different settings to determine speech intelligibility in noise and binaural properties with bilateral or unilateral Bonebridge: the summation effect, the squelch effect, and the head shadow effect. The subjects were asked not to move their head during the test and a calibration of the perceived signals with a sound level meter (Volcraft, Schallpegelmessgerät 332 Datalogger) was performed. For group 1, in the summation setting speech and noise were presented frontally with the same speaker (S_0_N_0_); in the squelch setting the signal was frontal, and the noise was presented to one ear (S_0_N_90_); in the head shadow setting, the speech was presented to one side and the noise to the contralateral ear (S_90_N_−90_). For group 2, in the summation setting speech and noise were both presented from the front of the patient (S_0_N_0_); in the squelch setting speech was presented from the front and noise on the side affected by conductive hearing loss (S_0_N_90_); in the head shadow setting, speech was presented on the side affected by conductive hearing loss and noise on the better ear (S_90_N_−90_). The Matrix test was performed on both groups at two different times (time 0 and time 1) in order to compare the gain in terms of binaural benefits of each group:-Group 1. Time 0: tests were performed using only the BCI in the worse ear, without wearing the contralateral one for 1 week. In fact, in case of patients affected by bilateral hearing loss implanted with bilateral aids, the ear that underwent surgery first was the one that showed lower verbal perception, so the worse ear was chosen to be aided first. Time 1: tests were performed with bilateral BCI.-Group 2. Time 0: tests were performed without any BCI. Time 1: tests were performed after implantation of BCI.

The difference between the results obtained at time 0 and those obtained at time 1 was calculated for each group in all three patterns and scores of the two groups were compared with each other to calculate the benefit in terms of binaural cues. Figure 1 shows ITA Matrix test performed in all three settings for both groups at time 0 and time 1.

Furthermore, each patient of both groups was administered a self-assessment questionnaire (APHAB—Abbreviated Profile of Hearing Aid Benefit—questionnaire), consisting of 24 questions to evaluate the perceived satisfaction of the subject itself with the bilateral or monaural BCI. The scores obtained provide the surgeon and the audioprothesist information regarding 4 sound properties under noise conditions: the ease of communication (EC), defined as communication under quiet conditions; the reverberation (RV), defined as communication under reverb conditions; the background noise (BN), defined as communication in places with different noise levels; the aversiveness (AV), defined as the discomfort deriving from environmental sounds. The difference between the GS obtained with bilateral or unilateral BCI resulted in the global benefit achieved by each patient from the implantation of the BCI. The subjects of the study were asked not to wear the BCI for 1 week and to fill in the questionnaire. Afterwards, they filled in the same questionnaire after at least 1 week of continuous use of the hearing aid. A global score (GS) obtained from the average of the four parameters’ scores for the two listening modes (without and with BCI) was calculated for each patient. All scores of APHAB questionnaire were expressed in the form of percentages. All surveys regarding APHAB questionnaire were carried out 2 months after BCI implantation for every subject of the study and all subjects used the implant correctly all day until our evaluation.

### 2.3. Statistical Analysis

Categorical variables are reported as frequency and percentage; continuous variables are reported as mean ± standard deviation (SD). Due to the small sample size of the study, the statistical analysis was performed using Mann–Whitney *U* test, a non-parametric test used to compare the means between two groups. The test allowed us to evaluate the significance of the difference between test results of patients belonging to group 1 (affected by bilateral conductive loss fitted with bilateral BCI) and patients belonging to group 2 (affected by monaural conductive hearing loss fitted with monaural BCI). The statistical significance was set at *p* < 0.05.

As for the APHAB questionnaire results were analysed using Mann–Whitney *U* test non-parametric test to highlight differences in subjective hearing and quality of life without and with BCI for all four categories (EC, RV, BN, AV) in the two groups of patients.

Statistical analysis was performed using IBM SPSS Statistics for Macintosh software, Version 28.0.

## 3. Results

Table 2 shows individual results of the Italian Matrix tests in all settings.

In the summation setting, patients of group 1 showed a mean SNR of 3.73 dB (SD, 7.77 dB) with unilateral BCI (time 0), compared with an average SNR of −0.93 dB (SD, 4.94 dB) with bilateral fitting (time 1). The difference between the results was 4.66 dB. Subjects of group 2 obtained a mean SNR of −2.11 dB (SD, 1.90 dB) without BCI (time 0) and a mean SNR of −2.90 dB (SD, 1.56 dB) with the BCI (time 1). The difference between results is 0.79 dB and resulted as statistically significant compared to the difference in SNR obtained by patients of group 1 (*p* < 0.05). In the squelch configuration, patients of group 1 obtained a mean SNR of −0.60 dB (SD, 8.15 dB) at time 0 and an average SNR of −2.84 (SD, 6.18 dB) at time 1. The difference between the results is 2.42 dB. Patients of group 2 showed a mean SNR of −3.06 dB (SD, 3.53 dB) at time 0 and an average SNR of −4.59 dB (SD, 3.89 dB) at time 1. The difference between obtained scores is 1.53 dB. In this setting, the differences obtained in the two groups resulted as not statistically significant (*p* = 0.85). In the head shadow setting, subjects of group 1 showed a mean SNR of 5.47 dB (SD, 7.57 dB) with unilateral BCI, compared with a mean SNR of −2.03 dB (SD, 2.60 dB) with bilateral BCI. The difference between time 0 and time 1 was 7.50 dB. Subjects of group 2 obtained an average SNR of 0.63 dB (SD, 2.58 dB) without BCI and an average SNR of −3.99 dB (SD, 3.76 dB) with the BCI. The difference between time 0 and time 1 was 4.61 dB. Also in this setting, the differences of the two groups resulted as not statistically significant (*p* = 0.34). Figure 2 shows the gain of benefit in all three settings for both groups.

Concerning the APHAB questionnaire in terms of GS, patients of group 1 reported a mean score of 33.00% (SD, 10.19%) at time 0 and of 21.90% (SD, 8.28%) at time 1, with a difference of 11.10%. Group 2 reported a mean score of 39.91% (SD, 15.27%) without BCI and of 43.92% (SD 12.09%) with the BCI; the gain between time 0 and time 1 was −4.00% for group 2, so showing a statistical significance as compared to the score obtained by group 1 (*p* < 0.05). All the items of the questionnaire resulted in a non-statistically significance of the differences between the two groups (*p* = 0.14 for the EC, *p* = 0.31 for the BN, *p* = 0.28 for the RV) except for the AV item, which showed a statistical significance in terms of subjective satisfaction between the two groups (*p* < 0.05). In particular, the AV scores’ average of group 1 was 21.29% (SD, 7.88%) at time 0 and 36.41% (SD, 24.78%) at time 1 with an increase in the perceived discomfort of 15.12% (and consequently a difference of −15.12% compared to the contralateral BCI). On the other hand, the AV score’s average of group 2 was 25.67% (SD, 16.70%) at time 0 and 69.00% (SD, 21.24%) at time 1, with a difference of −43.33%. Table 3 shows the individual results of the APHAB questionnaire.

## 4. Discussion

Over the years, many authors demonstrated that bilateral rehabilitation of patients affected by bilateral conductive hearing loss showed advantages in terms of binaural hearing leading to an improvement of speech perception under noise conditions, sound localisation and directional hearing [7,9,10,14,17]. On the other hand, binaural restoration of patients affected by monaural conductive hearing loss still remains a matter of debate among authors and its effectiveness in real-life hearing has not yet been certainly demonstrated [2,3,4,5]. In this work, we wanted to analyse and compare the efficacy in terms of binaural hearing achievement and subjective satisfaction of monaural BCI in patients affected by unilateral conductive hearing loss compared to bilateral BCI in patients affected by bilateral conductive hearing impairment, in order to understand the reasons behind opinion variability of previous authors when deciding whether to fit with unilateral BCI patients affected by monaural conductive hearing loss.

Scores obtained in our work lead us to make some considerations:-Our results reveal that patients of group 1 (affected by bilateral conductive or mixed hearing impairment fitted with bilateral BCI) started from lower hearing thresholds at baseline condition before treatment if compared to group 2 (affected by monaural conductive or mixed hearing loss fitted with unilateral BCI), resulting in a difference between group 1 and group 2 of 5.84 dB for the summation setting, 2.46 dB for the squelch setting and 4.84 dB for the head shadow setting at time 0. In fact, the implantation of a single BCI in subjects affected by bilateral conductive hearing loss results necessary in a disadvantaged starting condition if compared to subjects affected by unilateral hearing loss who received normal hearing input from the contralateral normal-hearing ear since their birth [18].-In our work, both groups showed a substantial decrement in the signal-to-noise ratio (SNR) at the speech recognition threshold (SRT) for all tested settings. However, group 2 obtained higher absolute scores in all analysed settings and, in contrast, a lower gain of benefit in terms of binaural cues if compared to group 1. There are several possible reasons that could explain these results. First of all, patients of group 2 were implanted when they were already adults and, as a result, they experienced binaural hearing deprivation for many years. Moreover, Stenfelt demonstrated that the so-called “transcranial attenuation” is responsible for affecting binaural hearing with a BC aid due to the additional stimulation of the cochlea contralateral to the BC device side [1].-According to Agterberg et al., who investigated binaural summation scores in patients affected either by bilateral conductive hearing loss implanted with bilateral BAHA or unilateral BC hearing loss fitted with unilateral BAHA, concluded that the former reported better scores in terms of binaural hearing [5]; our results account for the better speech perception in the noise of patients implanted with bilateral Bonebridge. This may be ascribed to the asymmetrical “aided” hearing thresholds obtained by patients of group 2 which also represents one of the reasons why differences obtained by the two groups showed statistical significance (*p* < 0.05) only in the summation setting. In fact, the different “starting condition” of group 1 and group 2 (which initially places group 2 at a distinct advantage over group 1 because of the normal-hearing ear) means that when signal and noise were presented both from the front, the gain of benefit obtained by group 1 with both BCI was much more consistent if compared to the gain of advantages obtained by fitting patients who already showed good ability to perceive sounds through to the non-aided ear (group 2). Indeed, bilateral BCI in patients affected by bilateral conductive hearing loss (group 1) led to symmetrical hearing thresholds and resulted in a great audiological improvement over the starting condition.-Patients of group 2 reported heterogeneous scores in terms of binaural benefits but showed a real binaural achievement also in the squelch setting, with a mean of −4.59 dB at time 1 and a difference of 1.53 dB if compared to the baseline situation. Since all patients in this group have been implanted in adulthood, we may assume that they experienced the same binaural hearing deprivation. For this reason, this heterogeneity in scores might be ascribed to the different individual TAs of each subject of the study.

Our audiometric results find some correspondence in the APHAB questionnaire which shows that patients of group 1 reported higher satisfaction with respect to their initial condition if compared to subjects of group 2, whose subjective performances seemed to be even better without than with the BCI (GS benefit: −4.00 dB). This suggests that, due to the different individual TA, the stimulation of the contralateral cochlea might cause disturbance in some patients affected by unilateral BC hearing loss fitted by unilateral BCI. Moreover, Priwin et al. found that some children with unilateral hearing loss used the BCI only in the classroom, thus revealing that unilateral BC hearing might show benefits in the school environment but seemed not to represent a subjective fundamental tool for patients in their everyday hearing life [2]. Agterberg et al. reported similar results and noticed that patients with bilateral conductive hearing impairment fitted with bilateral BCI showed subjectively better hearing when compared to subjects with unilateral hearing loss fitted with unilateral implants, especially in those patients who suffered from congenital unilateral hearing impairment [5].

## 5. Conclusions

In conclusion, despite the limits given by the low sample size, our data suggest that bilateral BCI in patients affected by bilateral conductive or mixed hearing loss might show more advantages in terms of speech perception in noise and subjective satisfaction if compared to unilateral BCI fitting in patients affected by unilateral conductive or mixed hearing impairment. This may be related both to the different transcranial attenuation of individuals which was considered to be responsible for the discomfort experienced by some patients fitted with unilateral BCI, and to the asymmetrical hearing thresholds obtained by such subjects. On the other hand, patients with unilateral conductive hearing loss fitted with monaural BCI achieved good results in terms of binaural hearing and for this reason, there is no absolute contraindication to implantation in these patients. Compared to the pre-existing literature, a pre-operative trial with softband or adhesive devices may be of value in patients affected by unilateral conductive hearing impairment in order to priorly select those who would benefit most from the procedure.

## Figures and Tables

**Figure 1 brainsci-13-01150-f001:**
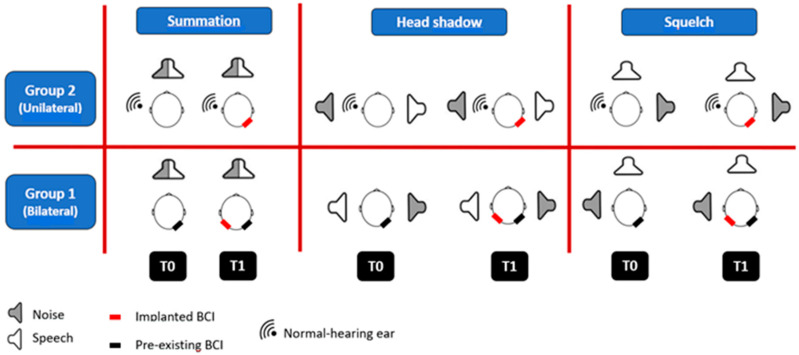
ITA Matrix test performed in all three settings for both groups at time 0 and time 1.

**Figure 2 brainsci-13-01150-f002:**
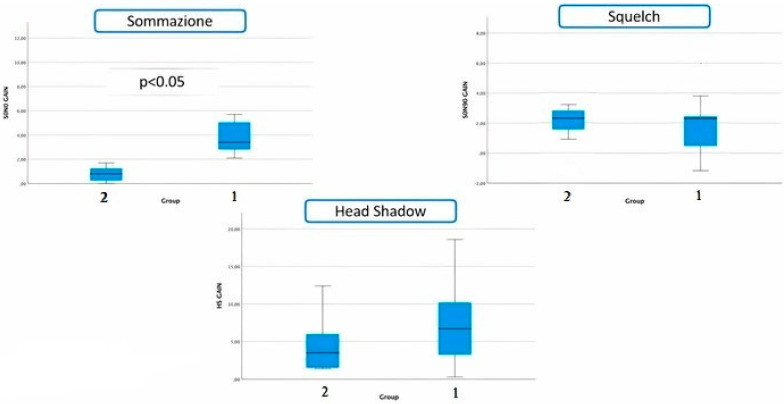
Gain of benefit of both groups for all three settings.

**Table 1 brainsci-13-01150-t001:** Anamnestic and baseline audiometric data of the samples. In this table, all the anamnestic characteristics (age at diagnosis, sex, type of hearing loss, side of hearing defect) and all the audiometric features at the main frequencies (0.5, 1, 2, 4 kHz) of the patients of each group are summarised.

**Group 1 (Bilateral Conductive Hearing Loss)**						**AC (BC) Thresholds (dB HL) at Frequency**			
**Patient**	**Age (yrs)**	**Sex**	**Congenital/Acquired**	**Side**	**Ear**	**0.5**	**1**	**2**	**4 kHz**
P1	29	F	Congenital	Bilateral	Right	85 (20)	70 (20)	60 (35)	60 (30)
					Left	85 (20)	70 (20)	70 (35)	75 (30)
P2	23	F	Congenital	Bilateral	Right	85 (20)	70 (30)	80 (50)	90 (50)
					Left	80 (20)	65 (30)	85 (50)	95 (50)
P3	9	F	Congenital	Bilateral	Right	65 (10)	65 (10)	55 (10)	55 (10)
					Left	65 (10)	70 (10)	55 (10)	50 (10)
P4	67	F	Acquired	Bilateral	Right	45 (20)	30 (20)	55 (30)	65 (40)
					Left	25 (20)	30 (20)	40 (30)	60 (40)
P5	16	F	Congenital	Bilateral	Right	90 (15)	75 (20)	75 (20)	70 (20)
					Left	80 (15)	85 (20)	80 (20)	70 (20)
P6	41	F	Congenital	Bilateral	Right	80 (20)	75 (25)	70 (30)	60 (25)
					Left	85 (20)	75 (25)	70 (30)	60 (25)
P7	20	F	Congenital	Bilateral	Right	80 (30)	95 (45)	85 (60)	85 (50)
					Left	80 (40)	100 (55)	90 (70)	90 (60)
**Group 2 (unilateral conductive hearing loss)**						**AC (BC) thresholds (dB HL) at frequency**			
**Patient**	**Age (yrs)**	**Sex**	**Congenital/Acquired**	**Side**	**Ear**	**0.5**	**1**	**2**	**4 kHz**
P1	68	M	Acquired	Left	Normal	30	30	35	35
					Impaired	70 (30)	70 (35)	85 (45)	85 (55)
P2	24	M	Congenital	Left	Normal	10	15	10	10
					Impaired	80 (15)	70 (15)	65 (20)	60 (25)
P3	61	M	Congenital	Right	Normal	15	15	20	30
					Impaired	85 (15)	60 (15)	75 (15)	65 (25)
P4	54	F	Acquired	Right	Normal	10	10	15	10
					Impaired	65 (30)	60 (20)	50 (35)	75 (35)
P5	43	F	Congenital	Right	Normal	10	10	10	10
					Impaired	80 (15)	75 (20)	60 (30)	55 (20)
P6	30	M	Acquired	Left	Normal	10	10	10	10
					Impaired	50 (15)	45 (10)	40 (15)	60 (15)
P7	56	F	Acquired	Left	Normal	25	30	20	25
					Impaired	55 (15)	65 (20)	70 (35)	50 (30)

Abbreviations: AC, air conduction; BC, bone conduction; dB HL, decibel hearing loss; kHz, kilohertz.

**Table 2 brainsci-13-01150-t002:** Results of Matrix Test in the three settings for both groups at T0 and T1. In this table, all the results obtained with Matrix Text in all settings at T0 and T1 are reported. As clarified in the “Materials and Methods” section, group 1 at T0 was fitted only with the BCI implanted in the worse ear, without using the contralateral aid.

	**Group 1 (Bilateral Conductive Hearing Loss)**					
	**Summation (dB)**		**Squelch (dB)**		**Head Shadow (dB)**	
	**T0** **(with one BCI)**	**T1** **(with two BCI)**	**T0** **(with one BCI)**	**T1** **(with two BCI)**	**T0** **(with one BCI)**	**T1** **(with two BCI)**
P1	2.6	0.5	1.2	−1.2	1	−3.9
P2	1.2	−4.5	−0.5	−3.7	7.2	−5.3
P3	3.4	−1.0	−4.2	−6.4	4.5	−3.4
P4	0	−3.3	−3.2	−5.5	0	−1.6
P5	−0.2	−3.6	−7.8	−6.4	4.9	−1.8
P6	−1.8	−4.1	−6.2	−7.1	−0.7	−1
P7	20.9	9.5	16.5	10.4	21.4	2.8
	3.73	−0.93	−0.60	−2.84	5.47	−2.03
	**Group 2 (unilateral conductive hearing loss)**					
	**Summation (dB)**		**Squelch (dB)**		**Head Shadow (dB)**	
	**T0** **(without BCI)**	**T1** **(with BCI)**	**T0** **(without BCI)**	**T1** **(with BCI)**	**T0** **(without BCI)**	**T1** **(with BCI)**
P1	−1.3	−3	−2.5	−1.9	3.4	−9
P2	−3.7	−4.7	−2.4	−6.2	−0.9	−5.9
P3	−0.6	−1.4	−4.6	−6.1	4.7	3.2
P4	−2.6	−2.8	−0.8	−3.1	−1.6	−3.1
P5	−1.8	−2.1	−3.3	−2.1	1.4	−5.6
P6	−5.2	−5.2	−9.6	−12	−1.7	−3.1
P7	0.4	−1.1	1.8	−0.7	−0.9	−4.4
	−2.11	−2.90	−3.06	−4.59	0.63	−3.99

Abbreviations: dB, decibel; BCI, bone conduction implant; T0, time 0; T1, time 1.

**Table 3 brainsci-13-01150-t003:** Individual results of the APHAB (Abbreviated Profile of Hearing Aid Benefit) questionnaire of both groups at T0 and T1. In this table, all the scores obtained in the APHAB questionnaire at each scale (EC, BN, RV, AV scale) and the global score of each group are reported at T0 and T1.

	**Group 1 (Bilateral Conductive Hearing Loss)**									
	**T0—with One BCI (%)**					**T1—with Two BCI (%)**				
	**EC Scale**	**BN Scale**	**RV Scale**	**AV Scale**	**GS**	**EC Scale**	**BN Scale**	**RV Scale**	**AV Scale**	**GS**
P1	19.17	37.67	12.83	33.50	25.79	1.00	37.67	12.83	48.00	24.88
P2	10.17	12.67	22.50	15.17	15.30	1.00	2.83	25.33	15.17	11.08
P3	41.50	46.00	54.17	12.83	38.63	6.50	20.83	10.17	23.00	15.13
P4	33.67	39.33	62.33	25.00	40.08	1.00	19.17	35.17	23.00	19.58
P5	33.17	60.33	54.00	24.83	43.08	12.33	24.83	20.67	87.00	36.21
P6	31.00	33.17	35.33	12.67	28.04	15.33	23.17	33.17	35.67	26.83
P7	33.67	39.33	62.33	25.00	40.08	1.00	19.17	35.17	23.00	19.58
	**Group 2 (unilateral conductive hearing loss)**									
	**T0—without BCI (%)**					**T1—with BCI (%)**				
	**EC Scale**	**BN Scale**	**RV Scale**	**AV Scale**	**GS**	**EC Scale**	**BN Scale**	**RV Scale**	**AV Scale**	**GS**
P1	41.67	45.83	49.83	24.50	40.46	6.5	47.67	43.33	83.00	45.13
P2	6.83	45.67	14.50	31.00	24.50	8.33	58.33	20.83	72.50	40.00
P3	82.83	51.67	56.17	60.17	62.71	83.00	49.67	58.17	64.17	63.75
P4	31.17	84.67	55.83	21.00	48.17	14.50	71.83	54.00	48.00	47.08
P5	6.50	29.00	15.75	18.50	17.44	37.50	29.00	22.00	87.00	43.88
P6	37.67	49.67	56.17	9.17	38.17	29.00	20.83	37.17	93.00	45.00
P7	23.33	70.50	82.67	15.33	47.96	1.00	28.83	25.17	35.33	22.58

Abbreviations: BCI, bone conduction implant; EC, ease of communication; RV, reverberation; BN, background noise; AV, aversiveness; GS, global score; T0, time 0; T1, time 1.

## Data Availability

Data Availability Statements are available in the section “MDPI Research Data Policies” at https://www.mdpi.com/ethics (accessed on 1 March 2022).

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
