# Peer review of "Comparison of Active Bone Conduction Hearing Implant Systems in Unilateral and Bilateral Conductive or Mixed Hearing Loss"

_brainsci, 2023, doi:10.3390/brainsci13081150_

Round 1

Reviewer 1 Report

This study evaluated the efficacy of active bone conductive hearing implants for patients with unilateral and bilateral hearing loss. The results showed that bilateral bone conduction implants in patients with bilateral conductive hearing loss have more advantages in terms of sound localization and speech perception compared to unilateral bone conduction implant in patients with unilateral conductive hearing loss. The number of cases is not so large, however the basic data in this study will be valuable for audiologists.

Author Response

Kind reviewer,

thank you for your interest in reading our work, we really appreciated your comment.

Our intention for the future is to expand the study sample so that we can obtain more accurate and validable results.

Thank you again,

the corresponding author

A. Canale

Reviewer 2 Report

The present manuscript assesses and compares binaural benefits and subjective satisfaction of active bone conduction implant (BCI) in patients with bilateral conductive or mixed hearing loss (HL) fitted with bilateral (BCI) and patients with monaural conductive HL fitted with monaural BCI.

Despite strong discussions over the years between researchers about its effectiveness and benefit as still lined with criticality, the authors might increase the sample size and ameliorate the result heterogeneity in scores of each subject of the study in order to priorly select the patients affected by unilateral conductive hearing impairment and finally those who would  benefit most from the procedure.

The language is in general correct.

Author Response

Kind reviewer,

thank you for your interest in reading our article, we really appreciated your comment.

Our intention for the future is certainly to expand the study sample so that we can have more accurate and validable results.

Thank you again,

The corresponding author

A. Canale

Reviewer 3 Report

The authors tried to demonstrate the benefit of bone conduction hearing aids in bilateral and unilateral hearing loss with audiological evaluations and a questionnaire study.

Title: It is not clear, I think "Comparison of active bone conduction hearing implant systems in unilateral and bilateral conductive or mixed hearing loss" can be in this way.

Introduction: ok

Method: In this section, in the patients section, it is written that group 1 consists of patients with bilateral hearing loss who have a monaaural bonebridge, and group 2 is composed of patients with unilateral hearing loss and a bonebridge inserted in that ear. However, in the later parts of the methods section, the use of bilateral bonebridge is mentioned in the section where the ITA matrix test is performed. And in the discussion section, I understand that a bilateral bonebridge is attached to group 1. This needs to be clarified. Group 1, which I understand from Figure 1, is that in addition to the pre-existing bonebridge, the second bonebridge was attached later, but you could not express it in the text. In summary there is a contradiction in what exactly was done to group 1.

  3. On page 110, tc should be corrected as ct.

Result: I think it would be helpful to simplified the tables in this section as much as possible and to write a short text summarizing all the tables.

Discussion: ok

Kind regards

Author Response

Kind reviewer,

thank you for your comment. We reviewed our article based on your suggestions and highlighted in yellow the changes made.

In particular:

- Title: It is not clear, I think "Comparison of active bone conduction hearing implant systems in unilateral and bilateral conductive or mixed hearing loss" can be in this way.We approved your suggestion regarding the title, that we changed as you proposed;

Thank you for your suggested title. We accepted you suggestion and changed or title.

- Method: In this section, in the patients section, it is written that group 1 consists of patients with bilateral hearing loss who have a monaaural bonebridge, and group 2 is composed of patients with unilateral hearing loss and a bonebridge inserted in that ear. However, in the later parts of the methods section, the use of bilateral bonebridge is mentioned in the section where the ITA matrix test is performed. And in the discussion section, I understand that a bilateral bonebridge is attached to group 1. This needs to be clarified. Group 1, which I understand from Figure 1, is that in addition to the pre-existing bonebridge, the second bonebridge was attached later, but you could not express it in the text. In summary there is a contradiction in what exactly was done to group 1.

We clarified audiological features of group 1 so that the comparison between the two groups is more understandable

- On page 110, tc should be corrected as ct.

On line 110, we changed TC with CT

- Result: I think it would be helpful to simplified the tables in this section as much as possible and to write a short text summarizing all the tables.

We checked the tables and added a short caption to each of them.

Thank you again for helping to improve the manuscript,

Best regards

The corresponding author

A. Canale

Round 2

Reviewer 2 Report

Published in the revised form.

Reviewer 3 Report

When I examine the text of the last revision and the answers to the criticisms, I see that the authors have corrected the errors in the article within the framework of the criticism. Acceptance of the article is appropriate, with the editor's discretion with the latest corrections.

Kind regards